# Evaluation of Safety and Immunogenicity of BNT162B2 mRNA COVID-19 Vaccine in IBD Pediatric Population with Distinct Immune Suppressive Regimens

**DOI:** 10.3390/vaccines10071109

**Published:** 2022-07-11

**Authors:** Nicola Cotugno, Enrica Franzese, Giulia Angelino, Donato Amodio, Erminia Francesca Romeo, Francesca Rea, Simona Faraci, Renato Tambucci, Elisa Profeti, Emma Concetta Manno, Veronica Santilli, Gioacchino Andrea Rotulo, Chiara Pighi, Chiara Medri, Elena Morrocchi, Luna Colagrossi, Giuseppe Rubens Pascucci, Diletta Valentini, Alberto Villani, Paolo Rossi, Paola De Angelis, Paolo Palma

**Affiliations:** 1Research Unit of Clinical Immunology and Vaccinology, Academic Department of Pediatrics (DPUO), Bambino Gesù Children’s Hospital, IRCCS, 00165 Rome, Italy; nicola.cotugno@opbg.net (N.C.); donato.amodio@opbg.net (D.A.); emmaconcetta.manno@opbg.net (E.C.M.); veronica.santilli@opbg.net (V.S.); gandrea.rotulo@opbg.net (G.A.R.); chiara.pighi@opbg.net (C.P.); chiara.medri@opbg.net (C.M.); elena.morrocchi@opbg.net (E.M.); grubens.pascucci@opbg.net (G.R.P.); 2Chair of Pediatrics, Department of Systems Medicine, University of Rome “Tor Vergata”, 00185 Rome, Italy; alberto.villani@opbg.net (A.V.); paolo.rossi@opbg.net (P.R.); 3The School of Pediatrics, University of Rome “Tor Vergata”, 00133 Rome, Italy; enrica.franzese@opbg.net (E.F.); elisa.profeti@opbg.net (E.P.); 4Digestive Endoscopy and Surgery Unit, Bambino Gesù Children’s Hospital, IRCCS, 00165 Rome, Italy; giulia.angelino@opbg.net (G.A.); erminia.romeo@gmail.com (E.F.R.); francesca.rea@opbg.net (F.R.); simona.faraci@opbg.net (S.F.); renato.tambucci@opbg.net (R.T.); paola.deangelis@opbg.net (P.D.A.); 5Department of Neuroscience, Rehabilitation, Ophthalmology, Genetics, Maternal and Child Health (DINOGMI), University of Genoa, 16126 Genoa, Italy; 6Microbiology and Diagnostic Immunology Unit, Bambino Gesù Children’s Hospital, IRCCS, 00165 Rome, Italy; luna.colagrossi@opbg.net; 7Pediatric Unit, Pediatric Emergency Department (DEA), Bambino Gesù Children’s Hospital, IRCCS, 00165 Rome, Italy; diletta.valentini@opbg.net; 8Academic Department of Pediatrics (DPUO), Bambino Gesù Children’s Hospital, IRCCS, 00165 Rome, Italy

**Keywords:** COVID-19 vaccine, IBD, pediatric

## Abstract

Patients affected by Inflammatory Bowel Disease (IBD) present higher risk for infection and suboptimal response upon vaccination. The immunogenicity of SARS-CoV2 vaccination is still largely unknown in adolescents or young adults affected by IBD (pIBD). We investigated the safety and immunogenicity of the BNT162B2 mRNA COVID-19 vaccine in 27 pIBD, as compared to 30 healthy controls (HC). Immunogenicity was measured by anti-SARS-CoV2 IgG (anti-S and anti-trim Ab) before vaccination, after 21 days (T21) and 7 days after the second dose (T28). The safety profile was investigated by close monitoring and self-reported adverse events. Vaccination was well tolerated, and short-term adverse events reported were only mild to moderate. Three out of twenty-seven patients showed IBD flare after vaccination, but no causal relationship could be established. Overall, pIBD showed a good humoral response upon vaccination compared to HC; however, pIBD on anti-TNFα treatment showed lower anti-S Ab titers compared to patients receiving other immune-suppressive regimens (*p* = 0.0413 at first dose and *p* = 0.0301 at second dose). These data show that pIBD present a good safety and immunogenicity profile following SARS-CoV-2 mRNA vaccination. Additional studies on the impact of specific immune-suppressive regimens, such as anti TNFα, on immunogenicity should be further investigated on larger cohorts.

## 1. Introduction

SARS-CoV-2 vaccinations with mRNA compounds largely proved their safety and efficacy in the healthy population. However, immune-compromised patients were initially excluded from vaccine trials.

Inflammatory bowel disease (IBD) is an immune-mediated inflammatory disease comprehensive of two major disorders: Ulcerative Colitis (UC) and Crohn disease (CD). The prevalence of IBD has been rising globally [1], with 20–30% of cases presenting before the age of 20 [2]. Whereas the safety of several vaccinations was proved in IBD patients [3], immunogenicity toward hepatitis B, hepatitis A, pneumococcal, and influenza vaccination was variably affected according to immunosuppressive therapies [4,5,6,7,8,9,10,11,12]. The latest evidence on SARS-CoV-2 vaccination immunogenicity revealed that IBD adult patients treated with the anti-tumor necrosis factor α (anti-TNFα) drugs (Infliximab (Janssen, Horsham, PA, USA), or Adalimumab (Abbvie, North Chicago, IL, USA)) developed a reduced immune response to SARS-CoV-2 vaccine, either BNT162b2 (Pfizer-Biontech, Mainz, Germany) or mRNA-1273 (Moderna, Cambridge, MA, USA) or ChAdOx1 nCoV-19 (Oxford–AstraZeneca, Cambridge, UK) [13,14,15,16,17,18,19] compared to other immune-suppressive regimens or age-matched healthy controls (HC).

Only recently, the SARS-CoV-2 mRNA vaccination was approved in the pediatric population after safety and immunogenicity studies were released [20]. However, despite the increasing extent of SARS-CoV-2 vaccination among children, little is known about safety and efficacy in children and adolescents with IBD (pIBD) [21,22] who are at increased risk for SARS-CoV-2 infection and SARS-CoV-2-induced disease relapses [23].

The members of the Porto group (the European Society for Pediatric Gastroenterology, Hepatology and Nutrition (ESP- GHAN), inflammatory bowel diseases (IBD) working group) recommend SARS-CoV-2 vaccination for children and adolescents with IBD and strongly suggested post-vaccination surveillance to monitor vaccine’s efficacy, side effects, and IBD disease course [24].

In the present study, we first aimed to confirm the safety of a two-dose SARS-CoV-2 mRNA vaccination schedule in adolescents and young adults with IBD (pIBD). We further investigated SARS-CoV-2-specific antibodies at baseline, 21 days after priming, and 7 days after the second dose. Additional markers of suboptimal immune response upon vaccination, such as the impact of anti-TNFα drugs in comparison with other immune-suppressive regimens, were evaluated.

## 2. Materials and Methods

### 2.1. Study Participants

Twenty-seven adolescents or young adults with IBD (pIBD) were enrolled between 12 March and 29 November 2021, at Bambino Gesù Children’s Hospital. In the present study, longitudinal blood samples were collected at the time of the first dose (T0), 21 days after first dose (T21), at the time of the second dose, and 7 days after second dose (T28). All patients were naïve to SARS-CoV2 infections as demonstrated by the absence of SARS-CoV-2 N-antibodies and no history of COVID-19; they all received BNT162b2 mRNA COVID-19 vaccine, with a schedule of 2 doses of 30 mcg 21 days apart [25]. Health-care workers who received BNT162b2 mRNA COVID-19 vaccine were used as control group (HC). All participants received a questionnaire about the adverse events and side effects following each dose of vaccine. All procedures performed in the study were in accordance with the ethical standards of the institutional research committee and with the 1964 Helsinki declaration and its later amendments. Local ethical committee approved the study and written informed consent was obtained from all participants or legal guardians. Age, gender, and clinical characteristics of the cohorts are described in Table 1.

### 2.2. Safety

Specific local or systemic adverse events and use of antipyretic or pain medication within 7 days of the receipt of each dose of vaccine were collected. Information was collected through a paper questionnaire reporting both solicited local and systemic adverse events and side effects following each dose of vaccine, up to 7 days after second dose. Events of IBD relapses were monitored during follow-up visits up to maximum 11 months after vaccination. Among the HC, no hospitalization occurred and only transitory adverse reactions in accordance with Polack et al. [26] were reported to the unit of occupational medicine by the employees enrolled.

### 2.3. Sample Collection and Storage

Venous blood was collected in EDTA tubes and processed within 2 h. Plasma was isolated from blood and stored at −80 °C.

### 2.4. Humoral Response

Anti-SARS-CoV2 IgG Antibodies (Ab) titers were measured as previously described [27] at T0, T21 and T28. In particular, we measured Ab against the S1-receptor-binding-domain (RBD) (Roche (Basel, Switzerland) cut-off: 0.8 U/mL) and anti-trimeric SARS-CoV-2 Ab (LIASION^®^ SARS-C0V-2 DiaSorin (Stillwater, MN, USA), cut-off: 13 AU/mL).

### 2.5. Quantification and Statistical Analysis

Statistical analyses were performed using GraphPad Prism 8 (GraphPad Software, Inc., San Diego, CA, USA). Statistical significance was set at *p* < 0.05 and the tests were two-tailed. All data were analyzed by D’Agostino–Pearson to assess normality and homogeneity by Levene’s test [28]. All variables included in the analysis were not normally distributed. Statistical tests were chosen according to data distribution in terms of normality, skewness, and dysomogeneity of variances. In line with this, as indicated in the figure legends, non-parametric paired test (Wilcoxon Signed Rank Test) was used to assess differences between Ab load at the different time points, whereas non-parametric unpaired test (Mann–Whitney test) was used for comparison between pIBD and HC and also between pIBD on therapy with anti-TNFα and patients receiving other treatments.

## 3. Results

### 3.1. Patient Characteristics

Among the pIBD, the mean age is 15.7 years (range 12.25–20.2 years, with 7 patients older than 18 years) and 15 patients with CD and 12 with UC are included. The full list of medication at enrollment is listed in Table 1. At every follow-up visit (every 2–3 months) during the course of the pandemics, all patients performed nasopharyngeal swab (NPS) screening for SARS-CoV2 infection. No infections were detected during the follow-up period. Among the pIBD, 14 patients out of 27 were on therapy with anti-TNFα (mean age 15.6 years with agerange 12.5–20.2 years, and with 4 patients older than 18). No differences in terms of age were found between patients on anti-TNFα compared to the other group (data not shown). As shown in Table 1, age at vaccination was significantly higher in HC (*p* = 0.0001). No differences related to gender were found between the groups.

### 3.2. Safety of 2 Doses of BNT162B2 mRNA COVID-19 Vaccine

Among the pIBD cohort, mild-to-moderate pain at the injection site within 7 days after the first dose was reported in 59.2% of the patients and the most commonly reported solicited local reaction. Only 1 out of 27 patients reported injection-site itch or injection-site swelling in a different case. The proportion of participants reporting local reactions remained almost the same after the second dose (62.9%), with pain reported in 55.5% of the patients, itch or redness reported in one patient, and swelling reported in 7.4% of the group. In general, solicited local reactions were mostly mild to moderate in severity and resolved within 1 to 3 days. Systemic adverse events were reported in 40.7% of patients after both the first and second dose (Table 2). The majority of patients reported mild-to-moderate systemic reactions; however, only one patient reported severe cold-like symptoms and asthenia after the second dose. The most commonly reported systemic reactions after the first dose were cold-like symptoms and myalgia, described, respectively, in 22.2% and 25.9% of the patients. Instead, the most frequent systemic reaction after the second dose was asthenia, reported in 33.3% of the patients, and myalgia and headache, reported in 22.2% of the patients. Only one patient after the first and the second dose reported the self-administration of anti-inflammatory treatments. No patients were hospitalized due to adverse reactions.

We also monitored the IBD course recording relapses for a follow-up period of a maximum of 11 months after priming. Five patients out of twenty-seven presented a disease relapse following vaccination; however, among these, only three patients had no clear history of recurrent IBD flares before the vaccination.

A 12-year-old girl was diagnosed with CD in September 2020 and presented before vaccination one relapse, for which she started Azathioprine with good clinical response. She received the first dose of the BNT162B2 MRNA COVID-19 vaccine on 9 September 2021 and the second dose on 30 September 2021. At the follow-up visit in October, the patient showed clinical sign of disease relapse in the perianal region followed in the coming weeks by pain, perianal secretion, and low-grade fever, with absence of response to topical therapy. The endoscopic examination showed active proctitis, for which she started Infliximab, with good clinical response.

An 18-year-old boy was diagnosed with UC in April 2017 and with history of two previous relapses before the immunization. At the time of vaccination, he was under treatment induction with systemic corticosteroids for severe pan colitis with an initial good clinical response. A few days following both the first and second dose of the BNT162b2 mRNA COVID-19 vaccine, the patient presented a rapid worsening with bloody diarrhea that required a treatment intensification with systemic steroid increase with only partial improvement, making a therapy switch to adalimumab necessary.

A 13-year-old girl, diagnosed with CD in November 2020 and with no history of previous relapse on treatment with adalimumab, showed wall thickening of the terminal ileum, cecum, and right colon at sonography, one week after the second dose of the BNT162b2 mRNA COVID-19 vaccine. The endoscopic exam showed an ileo-cecal stenotic valve and evidence of terminal ileum sub stenosis that required surgical resection.

Moreover, two patients already known for previous history of recurrent relapses presented additional disease reactivation after immunization.

### 3.3. Serologic Response

At the time of vaccination, all patients tested negative for anti-N, anti-S, and anti-trimeric antibodies, thus, excluding previous immunization or infection with SARS-CoV-2.

Overall, as shown in Figure 1 both the pIBD and HC groups experienced a statistically significant increase in anti-SARS-CoV-2 Abs after both the first and second vaccination dose (*p* < 0.0001).

Comparison of humoral response in terms of anti-S Abs between HC and pIBD did not show a statistically significant difference at both T21 and T28. Conversely, Trim Ab titers were significantly higher at both T21 and T28 in pIBD compared to HC (*p* < 0.0001 and *p* = 0.0061, respectively) (Figure 1).

Further analyses were performed to understand whether therapy with anti-TNFα agents was associated to a lower Ab response. The comparison of anti-S Ab between patients on therapy with anti-TNFα and patients receiving other treatments showed that the anti-S Ab titers were significantly reduced in the anti-TNFα group after both the first and second dose (*p* = 0.0413 and *p* = 0.0301, respectively), whereas trim Ab titers were comparable for the two groups at both timepoints (Figure 2).

## 4. Discussion

In the present study, we report the safety profile and the immunogenicity of the BNT162B2 SARS-CoV2 ((Pfizer-Biontech, Mainz, Germany) in a cohort of adolescents and young adults with IBD and with no previous history of COVID-19.

Our data confirmed a good safety profile, with the majority of local and systemic adverse events being transient and ranging from mild to moderate, with no reported cases of hospitalization or additional treatment.

One of the major concerns regarding pIBD presenting with SARS-CoV2 infection is the higher risk of disease relapse, often caused by transitory interruption of the immune-suppressive treatments [23]. Through a long-term follow-up after vaccination, we here investigated whether SARS-CoV-2 vaccination could impact on the rate of IBD flares over time. According to what was previously found in adults, who showed no increased risk of short-term adverse effects [29,30,31,32] or IBD flares [29,31,32,33] after vaccination, only 5 patients out of 27 (18.5%) showed an IBD exacerbation after SARS-CoV-2 vaccination. However, among these patients, two already had a clinical history of recurrent disease relapse. Thus, no causal correlation between primary disease exacerbation and SARS-CoV-2 vaccination could be established in our cohort, highlighting the critical importance of additional studies characterized by a larger sample size of pIBD and longer follow-up.

In the context of the clinical management of chronic conditions, such as IBD, the pandemic worsened the follow-up of these patients, both in terms of timing of follow-up visits and in terms of therapeutic management in case of infection [23,34]. The advent of SARS-CoV-2 vaccination represents a crucial milestone in the history of pandemics, especially for patients with acquired immune deficiencies, due to ongoing immune-suppressive treatments. However, large hesitation from the IBD adult community towards the vaccine intervention for both safety and immunogenicity reasons was noted at the beginning of the vaccination campaign [29,35].

In addition to confirming a good safety profile, here, we showed a good effectiveness of the SARS-CoV-2 vaccination in an IBD pediatric population. The anti-S SARS-CoV-2 Ab response to BNT162B2 SARS-CoV-2 in our cohort of pIBD, naïve to SARS-CoV-2 infection, was similar compared to HC at 21 days after priming and 7 days after the second dose. Confirming the good immunogenicity profile, anti-Trim Abs were higher in pIBD compared to HC at both time points. Such a difference may be due to the older age of the HC group, mainly composed of healthcare workers, considering the previously demonstrated influence of age on SARS-CoV-2 vaccination response and long-term memory maintenance [36]. The independent impact of age on immunogenicity was also confirmed in multiple adult IBD cohorts [15,17] and by our group in another cohort of immune-compromised children [37].

In the present study, we further explored the impact of specific immune-suppressive regimens on the immunogenicity of BNT162B2 in pIBD. A suboptimal response in adult IBD on therapy with anti-TNFα in comparison with either IBD treated with other immune-suppressive regimens or HC was recently published [13,14,15,16,17,18]. For the first time in the pediatric context, we confirmed such findings, showing that pIBD on therapy with anti-TNFα (Infliximab or Adalimumab) developed a lower response in terms of Anti-S ab compared to patients receiving Azathioprine or Mesalazine. In contrast with these results, showing no effect of azathioprine on SARS-CoV-2 vaccination immunogenicity, the negative influence of another antimetabolite, the mycophenolate, was recently shown in a cohort of solid-organ-transplanted adolescents and young adults [38]. A separate analysis by Spencer et al. [22] showed a similar serologic response after vaccination or infection in IBD young adults treated with distinct biological treatments, such as adalimumab, ustekinumab, and tofacitinib.

In conclusion, our data support the safety and immunogenicity of the mRNA SARS-CoV-2 vaccination in pediatric IBD population. A long-term surveillance of serologic response and personalization of the booster dose in patients receiving anti-TNFα drugs may be considered. The major limitations of the present study were the small sample size and the absence of an age-matched control group for HC. Further studies, performed in larger cohorts of pIBD with an age-matched control group, a longer time of follow-up, and with additional markers of cell-related immunogenicity, should be considered.

## Figures and Tables

**Figure 1 vaccines-10-01109-f001:**
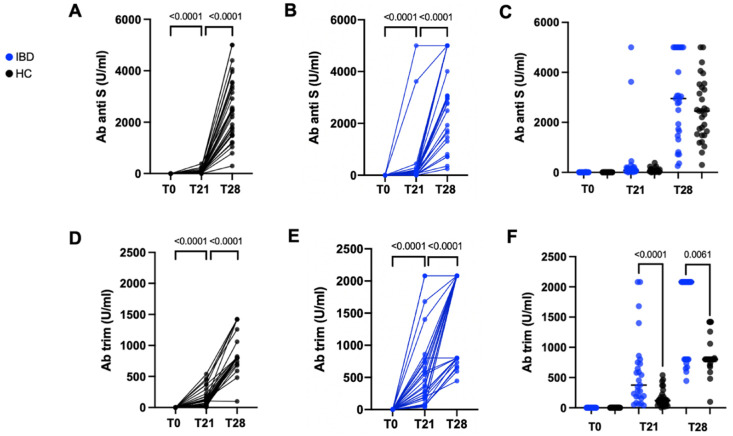
Longitudinal analysis of anti-S ab (**A**,**B**) and trim ab (**D**,**E**), respectively, in HC and pIBD. Paired non-parametric test (Wilcoxon Signed Rank Test) was performed to define longitudinal Ab increase (**A**,**B**,**D**,**E**). Differences between HC and IBD were calculated for anti-S Ab (**C**) and trim Ab (**F**) at T0, T21, and T28. Unpaired non-parametric test (Mann–Whitney test) was used for comparison.

**Figure 2 vaccines-10-01109-f002:**
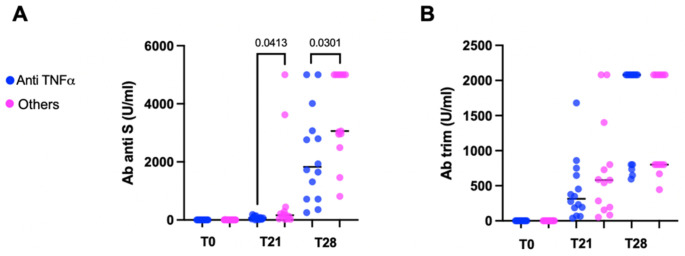
Difference between pIBD patients on anti-TNFα drugs and pIBD patients receiving other treatment regimens in terms of anti-S ab (**A**) and trim ab (**B**). Unpaired non-parametric test (Mann–Whitney test) was used for comparison.

**Table 1 vaccines-10-01109-t001:** Characteristics of the study cohort.

	**pIBD N = 27**	**HC N = 30**
**Females, N (%)**	8 (29.6)	21 (70)
**Age (years), Median (Range)**	15.7 (12.25–20.2)	36 (23.8–42.5)
**IBD Subtype, N (%)** Crohn’s disease Ulcerative Colitis	15 (55.6) 12 (44.4)	n.a.
**VEO-IBD, N (%)**	2 (7.4)	n.a.
**Crohn’s Disease Location, N (%)** Ileal Colonic Ileocolonic Isolated Upper Tract	3 (20) 1 (6.7) 11 (73.3) 0	n.a.
**Ulcerative Colitis, N (%)** Proctitis Left-sided Extensive/pancolitis	1 (6.7) 4 (26.7) 7 (46.7)	n.a.
**IBD Therapy, N (%)** IFX IFX + other immunosuppressant (Azathioprine, MTX) Adalimumab alone or Adalimumab+ Mesalazine Azathioprine alone or Azathioprine+Mesalazine Mesalazine Mesalazine + GC	2 (7.4) 4 (14.8) 8 (29.6) 4 (14.8) 8 (29.6) 1 (3.7)	n.a.

VEO-IBD = Very Early Onset IBD; IFX = inflix imab; MTX = methotrexate; GC = glucocorticoids.

**Table 2 vaccines-10-01109-t002:** Adverse reactions in pIBD cohort.

	**first Dose, N = 27**	**second Dose, N = 27**
**Local Adverse Events, N (%)**	16 (59.2)	17 (62.9)
Pain in the site of injection	16	15
Itch	1	1
Hyperemia	0	1
Edema	1	2
**Systemic adverse events, N (%)**	11 (40.7)	11 (40.7)
Fever	0	2
Cold-like symptoms	6	5
Asthenia	5	9
Myalgia	7	6
Headache	4	6
Pharyngodynia	3	1
Vomit	0	3
Lymphadenopathy	0	1
**Self administered antiinflammatory or antipyretic treatment**	1	1

## Data Availability

Data generated over the study and supporting the results presented in the paper will be provided upon request.

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
