# Peer review of "Evaluation of Safety and Immunogenicity of BNT162B2 mRNA COVID-19 Vaccine in IBD Pediatric Population with Distinct Immune Suppressive Regimens"

_vaccines, 2022, doi:10.3390/vaccines10071109_

Round 1
Reviewer 1 Report
Main comments: In this study the authors investigated safety and immunogenicity of 2 doses of BNT162B2 mRNA vaccine in children or young adults (n=27) with inflammatory bowel disease compared to adult healthy controls (n=30). The authors found a good safety and humoral immune response.
Overall, this is an interesting clinical study. However, I recommend to correct/clarify the following points:
- The authors are imprecise about the age group of their patients. According to the title (line 3, line 226, line 250) and abstract (line 27), they investigated "pediatric population" or 27 "children". This is not completely true, as according to Table 1, the 3. quartile of age in patients was 20.2 years. Therefore, the authors need to clarify the proportion of adults (>=18 years) and they should give the minimum age of patients (12 years?) and the total range.
- As for basic immunization 2 doses of BNT162B2 are required (also in children), the authors should not call the second dose a "booster" but as it is a "second dose" (line 29, line 79)
- Adverse reactions in healthy controls were not reported, therefore they need to be added as comparison in Table 2.
- According to Figure 1 and 2 the authors used "non parametric T tests". As the T test is always a parametric test, this is not possible.
- Throughout the whole manuscript, for all numbers with decimal points the authors should use a point and not a comma.
Minor comments:
Abstract:
-line 27: clarify how many of the 27 patients were really children
-line 28: change "booster dose" to "second dose"
Material and Methods
-line 76: change "twenty-seven children and young adults" to "twenty-seven children or young adults" and define the minimum age of included patients.
-line 79: do not use the term booster dose
-Table 1: include abbreviations as footnotes
Results
-line 118: here you show the interquartile range but not the "total" range, however the total range would be interesting
-Table 2: please include adverse reactions of healthy adults
-Figure 1: For IBD patients also show lines in blue. and clarify if you performed a non-parametric or a parametric test for comparisons
-Figure 2: The same concerns regarding the statistical test. Please also give the age groups of the subgroup of patients on anti-TNF alpha versus other treatment regimens.
Reviewer 2 Report
Authors summarized clinical experimentation with pediatric patients affected by IBD.
Before any further recommendation can be given, the following principal questions should be responded.
Major comment 1: The design of the study is insufficient. trial and compare groups are inconsistent, mean and median ages are very different, moreover,
there is no placebo balancing group involved at all. Thus nothing on effectiveness can be stated based on this experiment (see e.g. lines 224-232 in Discussion).
All p-values can be well caused by age-difference.
Major comment 2: Statistical analysis is insufficiently reported and performed.
1)
different sized trial and control often cause different variances. No homogeneity of variances testing is performed.
2) Moreover, in the normality test, if the sample size is small, the power is not guaranteed. p-values of directed normality tests are missing, and chose of p-value 0.05 put a cautionary question on p-hacking.
3) no independence pre-tests before application of t-tests are given.
Authors shall carefully respond to all 1,2,3), more reading for necessity of doing proper mean-comparisons is
More about the basic assumptions of t-test: normality and sample size, by Tae Kyun Kim and Jae Hong Park
and https://doi.org/10.1080/03610918.2019.1649698
Unfortunately, serious issues should be fixed before any consideration on the manuscript can be given and current version is out of scope.
Scientific quality of the manuscript is both low and misleading in its current form.
The topic itself may be potentially interesting, but only after serious revisions.
Improvements, both in explanation, interpretation of method, illustration and very careful justification of precision of all used methods is needed.
Round 2
Reviewer 1 Report
In the revised version of the manuscript the authors did consider all reviewer comments. But I still recommend to transfer the information about safety in healthy controls (line 101-103) from materials and methods to the results.
Author Response
We thank the reviewer for acknowledging the improvement that have been made to the current version of the manuscript following his suggestions.
Following reviewer's suggestion, we have now edited the manuscript for minor typos.
on behalf of All Authors,
Paolo Palma, MD, PhD
Reviewer 2 Report
Dear Authors: Unfortunately, despite some efforts authors did not solved the outlined serious problems.
First, homogeneity analysis for variances was not performed, this is necessary pre-test before any means comparison can be valid.
Secondly, the low p-values for normality null hypothesis computed by D'Agostino-Pearson test (published in Table of answers) are caused by serious skewness in the data. These skewnesses are well depicted also in the data plots.
But, the Wilcoxon-Mann-Whitney test is invalid when discrete and/or extremely skew data are analyzed.
Thus before any positive recommendation can be given, these flaws should be checked carefully, i.e. both homogeneity of variances and skewness of the data-sets. Sincerely, Referee
Round 3
Reviewer 2 Report
Authors did some work but it is still not satisfactory for making a publication recommendation.
Major comment 1:
First, F test for homogeneity of variances has a prerequisite of normal distribution, which is not satisfied.
Major comment 2: Authors shall compute skewness and present it to table, in order to clarify whether the low p-values for normality null hypothesis computed by D'Agostino-Pearson test (published in Table of answers) are caused by serious skewness in some of the distributions. But, the Wilcoxon-Mann-Whitney test is invalid when discrete and/or extremely skew data are analyzed.
Referring to Fay MP, Proschan MA is missing justification, since they considered Wilcoxon-Mann-Whitney for distributions with heavy tails or very skewed distributions. But medium skewed may fail as well. So more analysis is needed.
For Happ M, Bathke AC, Brunner E. we need a better sampling design for the Wilcoxon-Mann-Whitney test, in order to have a good performance.
Sincerely, Referee
Round 4
Reviewer 2 Report
Authors did some work but it is still not satisfactory for making a publication recommendation.
On the Major comment 1:
Authors replaced F test for homogeneity by test of Levene. However, Levene test also requires as prerequisite normality,
, which is not satisfied (it uses F-distribution by p-value evaluation).
This shall be fixed by using homogeneity test which is distribution free.
Second, even if we wish to consider Levene, we received heterogeneity. So we cannot conclude statistically correctly difference in immuno-response.
If we do not have homogeneous variances, we cannot compare means!
On the Major comment 2:
Authors computed skewness and present it to table, and Referee needs to conclude that skewness larger than 1.1 is not compatible with correct usage of
Wilcoxon-Mann-Whitney. Such high levels of skewness is reflecting in such small sample sizes distributions with heavy tails. Thus, results of Wilcoxon-Mann-Whitney are not justified and more careful analysis is needed.
Author Response
Response submitted to the Editor.